# Neuronal Dynamics of Pain in Parkinson’s Disease

**DOI:** 10.3390/brainsci11091224

**Published:** 2021-09-16

**Authors:** Kaoru Kinugawa, Tomoo Mano, Kazuma Sugie

**Affiliations:** Department of Neurology, Nara Medical University, Kashihara, Nara 634-8521, Japan; kinugawa_kaoru@naramed-u.ac.jp (K.K.); ksugie@naramed-u.ac.jp (K.S.)

**Keywords:** pain, Parkinson’s disease, electroencephalography, neuronal synchronization

## Abstract

Pain is an important non-motor symptom of Parkinson’s disease (PD). It negatively impacts the quality of life. However, the pathophysiological mechanisms underlying pain in PD remain to be elucidated. This study sought to use electroencephalographic (EEG) coherence analysis to compare neuronal synchronization in neuronal networks between patients with PD, with and without pain. Twenty-four patients with sporadic PD were evaluated for the presence of pain. Time-frequency and coherence analyses were performed on their EEG data. Whole-brain and regional coherence were calculated and compared between pain-positive and pain-negative patients. There was no significant difference in the whole-brain coherence between the pain-positive and pain-negative groups. However, temporal–temporal coherence differed significantly between the two groups (*p* = 0.031). Our findings indicate that aberrant synchronization of inter-temporal regions is involved in PD-related pain. This will further our understanding of the mechanisms underlying pain in PD.

## 1. Introduction

Parkinson’s disease (PD) is a well-known neurodegenerative disease. It is characterized by motor and non-motor symptoms (NMSs) [1,2]. Dopaminergic medications and deep brain stimulation therapy improve the motor symptoms and quality of life of patients with PD. However, NMSs are frequently difficult to treat. Many patients with PD experience pain as an NMS, which negatively impacts their quality of life [3]. PD-related pain never responds to analgesic medications, although it sometimes responds to dopaminergic medications [4]. Moreover, the mechanisms underlying pain in PD have not been sufficiently studied [5].

Pain in patients with PD may result from altered sensory processing [6]. Previous studies found that, as seen in central sensitization, abnormal neuronal synchronization occurs in the brain networks of patients with PD with pain [7]. Electroencephalography (EEG) is a simple method for measuring brain neuronal activity and synchronization [8]. Quantitative EEG analysis provides information about whole-brain and regional connectivity [8]. EEG coherence has been used as a biomarker for the clinical severity of PD [9].

Previous studies on neuronal synchronization have compared patients with PD with healthy controls. However, a comparison between patients with PD with and without pain has not yet been performed. Thus, our study aimed to elucidate the association between neuronal synchronization in neuronal networks and PD-related pain, using EEG coherence analysis. To this end, we investigated the difference in coherence between patients with PD, with and without pain.

## 2. Materials and Methods

### 2.1. Patients and Clinical Data

We included patients with continuous sporadic PD who visited Nara Medical University Hospital, Nara, Japan. The inclusion criteria were (1) Japanese patients with confirmed PD who met the U.K. PD Society Brain Bank clinical diagnostic criteria [10], (2) patients aged 40–90 years, and (3) patients with no significant brain lesions on magnetic resonance imaging. The exclusion criteria were (1) patients receiving medications that could influence the EEG findings (e.g., antianxiety medication, antidepressant medication, such as serotonin–noradrenaline reuptake inhibitors and selective serotonin reuptake inhibitors, antiepileptic medication, and opioids), (2) patients who had undergone operations, such as deep brain stimulation, (3) patients with severe cognitive impairment (Mini-Mental State Examination score < 10), and (4) patients with other known causes of pain (e.g., orthopedic disease, peripheral neuropathy, spinal cord disease). Pain was assessed using the Movement Disorder Society Unified Parkinson’s Disease Rating Scale Part I (MDS-UPDRS-I) scores for the item of pain and other sensations (Q 1.9) [11]. This item was rated on a scale of 0–4: (0—normal: no uncomfortable feelings such as pain, aches, tingling, or cramps. 1—slight: I have these feelings. However, I can do things and be with other people without difficulty. 2—mild: These feelings cause some problems when I do things or am with other people. 3—moderate: These feelings cause a lot of problems, but they do not stop me from doing things or being with other people. 4—severe: These feelings stop me from doing things or being with other people.). EEG (Neurofax EEG-1224, Nihon Kohden, Tokyo, Japan) was performed, using the international 10–20 electrode placement system and a sampling rate of 200 Hz. The electrodes were referenced to linked earlobes (A1 + A2). During EEG recordings, patients were instructed to relax in a supine position and avoid blinking as much as possible. The EEG examinations and clinical assessment were carried out at the same time of day on separate days within one month with the patients in the “on” phase after 1 to 2 h, while on their usual medications. As such, we sought to exclude the cases of dystonia-related pain in the “off” phase. This study was approved by the Clinical Research Ethics Board of Nara Medical University. Informed consent was obtained from all patients. 

### 2.2. EEG Analysis

EEG data were analyzed, using MATLAB (version R2020a; Math Works) and EEGLAB toolbox (version 14.1.2) [12]. The EEG data were band-pass filtered, using finite-impulse response filtering. A low-pass filter of 45 Hz and high-pass filter of 1 Hz were applied. Since EEG data are often contaminated with artifacts at the beginning of a recording, we manually selected 60 s of resting-state eyes-closed EEG data 5 min after recording began. These data were segmented into 2 s epochs, and artifact-contaminated epochs were removed. Time-frequency analysis was performed across all 16 electrodes, using fast Fourier transformation with a wavelet transform and a frequency resolution of 1 Hz (Appendix A). The EEG data of the following four frequency bands were separately analyzed and compared between the groups: theta (4–7 Hz), alpha (8–12 Hz), beta (13–30 Hz), and gamma (31–45 Hz) [13]. The EEG data underwent notch filtering at 50 Hz and were decomposed, using wavelet transformation with two cycles per frequency band. A coherence analysis was performed between all electrode pairs [12]. All 16 electrodes were assigned to four brain regions (frontal—Fp1, Fp2, F3, F4, F7, and F8; parietal—C3, C4, P3, and P4; temporal—T3, T4, T5, and T6; and occipital—O1 and O2), and electrode pairs were categorized into 10 groups representing brain regions (frontal–frontal (FF); frontal–parietal (FP); frontal–temporal (FT); frontal–occipital (FO); parietal–parietal (PP); parietal–temporal (PT); parietal–occipital (PO); temporal–temporal (TT); temporal–occipital (TO); occipital–occipital (OO)). Whole-brain coherence was assessed by averaging the coherence values of all the electrode pairs, and regional coherence was assessed by averaging the coherence values of electrode pairs based on the region for each frequency band [14,15]. For example, the value of TT coherence was assessed by averaging the values of T3-T4, T3-T5, T3-T6, T4-T5, T4-T6, and T5-T6 [14]. The inter-regional coherence indicated FP, FT, FO, PT, PO, and TO coherence, and the intra-regional coherence indicated FF, PP, TT, and OO coherence. To evaluate inter-and intra-regional connectivity, regional coherence was compared between the pain-positive and pain-negative groups. We also probed the relationship between regional coherence and clinical characteristics in both groups.

### 2.3. Statistical Analyses 

The means of variables were compared between pain-positive and pain-negative patients with PD, using an independent two-sample t-test and chi-square test. A Shapiro–Wilk test was used to assess the distribution of the data. Statistical significance was set at *p* < 0.05. A correlation was considered strong if the correlation coefficient (*r*) was >0.40. Analyses were performed in SPSS 22.0 J (IBM Japan, Tokyo, Japan).

## 3. Results

Twenty-four patients with PD (14 men and 10 women) fulfilled the selection criteria and were enrolled in this study. The pain-positive and pain-negative groups comprised 12 patients (7 men and 5 women) each. The demographic characteristics, levodopa-equivalent daily dose, and clinical features of the 24 patients with PD are summarized in Table 1. There was no statistically significant difference between the two groups in these variables, which included key variables, such as age, disease duration, MDS-UPDRS-III score, levodopa-equivalent daily dose [16], and Hoehn–Yahr scores. The mean score of MDS-UPDRS-I for pain and other sensations was 2.25 ± 0.75. One patient (8.3%) had a score of 1 (slight), eight (55.7%) patients had a score of 2 (mild), two (16.7%) patients had a score of 3 (moderate) and one (8.3%) patient had a score of 4 (severe; Table 1). Four of the patients (33%) had continuous pain, and eight (67%) patients had discontinuous pain.

No significant differences were identified in whole-brain coherence in the alpha band between the pain-positive and pain-negative groups (*p* = 0.39). The coherence of the TT region in the alpha band significantly differed between the two groups (*p* = 0.031; Table 2) and was not correlated with the MDS-UPDRS-III score or disease duration in the pain-positive group (*r* = −0.053 and −0.048, respectively). However, these variables were correlated in the pain-negative group (*r* = −0.40 and 0.55, respectively; Figure 1). The pain-positive group was further divided into limb-pain (*n* = 5) and trunk-pain (*n* = 7) groups, and the whole-brain coherence, regional coherence, MDS-UPDRS-III score, and disease duration were compared between these two groups. The coherence of the TT region was higher in the trunk-pain group than in the limb-pain group (*p* = 0.034). There were no significant differences between the limb-pain and trunk-pain groups in the MDS-UPDRS-III score (*p* = 1.00) or disease duration (*p* = 0.40).

## 4. Discussion

Here, EEG coherence in the alpha frequency band, used to evaluate neuronal synchronization in brain networks, was associated with pain modulation. Functional abnormalities can result in the reorganization of the sensory system [17]. This can lead to central sensitization, defined as an increased responsiveness of nociceptive neurons in the central nervous system to normal or subthreshold afferent input [18]. Pain hypersensitivity is elicited by neural signal amplification. Central sensitization has been thought to play a role in unexplained pain in several disorders, such as fibromyalgia, epilepsy, Alzheimer’s disease, and PD [19,20]. Alongside limb pain, patients with PD commonly experience trunk pain, including abdominal pain [21]. Various factors, such as altered posture, muscle tone abnormalities, and truncal dystonia, contribute to trunk pain in patients with PD. However, trunk pain is frequently neglected and insufficiently treated in patients with PD [22]. 

An EEG coherence analysis revealed that patients with PD with pain showed higher regional coherence, representing increased functional connectivity, in the temporal regions. The temporal lobe is generally responsible for establishing long-term memory, cognitive and emotional functions, and auditory processing [23]. However, several neuroimaging studies have reported altered activation within this region [24,25,26]. In patients with chronic low back pain, the anterior hippocampus, a part of the medial temporal lobe, showed significantly lower levels of activity and functional connectivity than the medial prefrontal cortex [26]. The anterior hippocampus is involved in mood-related functions and psychological modulation, likely through interactions with the amygdala. The amygdala regulates psychological responses and has been thought to play a role in pain-related negative affect processing. The amygdala may induce hypoalgesia to modulate pain in individuals with psychological stress. Anterior hippocampus–amygdala interactions are known to be involved in both the encoding and retrieval of affective information, and this has been observed in individuals with experimental pain. Thus, pain-related abnormal anterior hippocampal activity may be related to psychological dysregulation. Pain may be considered a stressor, and it elicits a prolonged stress response; this implies that pain poses an allostatic load on the neuronal networks. The hippocampus is particularly sensitive to the neurotoxic effects of prolonged exposure to psychological stress, which affects its structure and function [26]. Our results are consistent with the previous findings, demonstrating that aberrant connectivity in the temporal lobe could result in pain. Moreover, we found that temporal lobe connectivity was not correlated with the MDS-UPDRS-III score or disease duration in patients with PD with pain, which is consistent with the findings of a previous study [27]. These results suggest that higher temporal lobe inter-regional connectivity is related to pain in patients with PD.

Our study sheds light on the possible mechanisms underlying PD-related pain; however, several limitations should be considered. First, we presented a novel approach to evaluate PD-related pain, using EEG coherence analysis. It may be key in elucidating the mechanism of pain in PD cases. This clinical study was a pilot study with a limited sample size [9]. Based on the obtained results, a large clinical trial should be performed to validate our findings in the appropriate populations. Second, our study focused on the difference in neuronal synchronization between patients with and without pain and did not consider other clinical factors that might influence pain, such as depression [28], which has been associated with abnormal connectivity in the default mode network in patients with PD [29]. Many other factors, such as age, sex, and medical history, may also contribute to the reorganization of functional connectivity in neuronal networks [30]. Third, we did not assess pain intensity, because the intensity of PD-related pain fluctuates throughout the day [31]. Additionally, to evaluate cortical–subcortical synchronization with pain, further studies, such as stereo-EEG and functional MRI studies, are needed [7,32].

## 5. Conclusions

In conclusion, we performed an EEG coherence analysis in patients with PD, with and without pain. Aberrant neuronal synchronization and abnormal inter-temporal lobe connectivity may be involved in PD-related pain. By performing the EEG analysis, we examined the mechanisms of abnormal connectivity underpinning pain in PD. There have been numerous basic and clinical research studies on pain in patients with PD; however, the mechanism remains unclear. Therefore, it may be useful to evaluate pain from a new perspective by performing a non-invasive neurophysiological technique.

## Figures and Tables

**Figure 1 brainsci-11-01224-f001:**
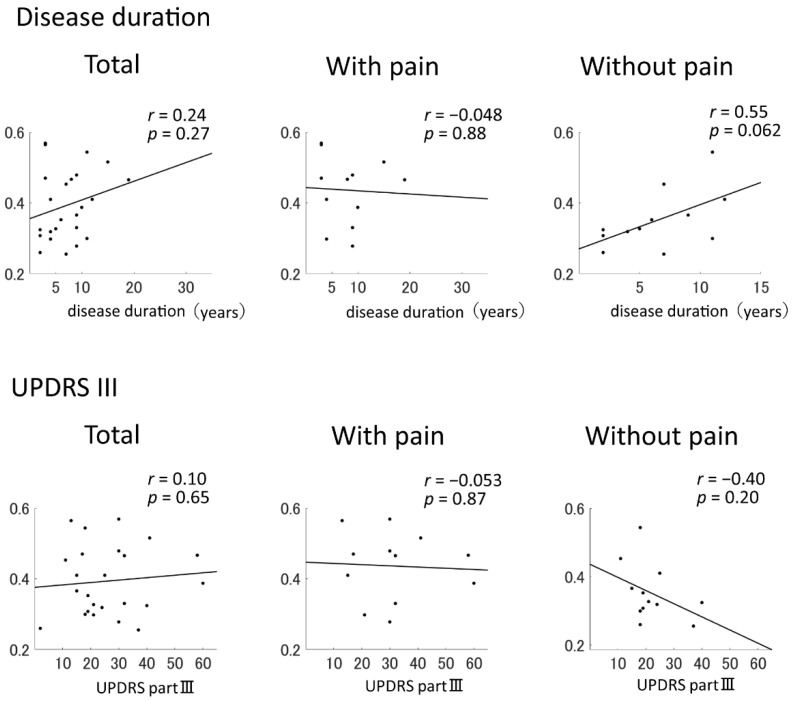
Pearson correlation coefficients and graphs for the correlation of temporal–temporal coherence in the alpha range with disease duration and MDS-UPDRS-III score. The lines indicate the approximate line using the least squares method. MDS-UPDRS-III = Movement Disorder Society Unified Parkinson’s Disease Rating Scale Part III.

**Table 1 brainsci-11-01224-t001:** Demographic and clinical characteristics of the participants.

	PD	PD with Pain	PD without Pain	*p* Value
Number	24	12	12	
Age	72.29 ± 8.70	72.58 ± 10.80	72.0 ± 6.42	0.87
Sex	14 Male/10 Female	7 Male/5 Female	7 Male/5 Female	1.00
Disease Duration, years	7.25 ± 4.40	8.08 ± 5.28	6.5 ± 3.66	0.40
Pain Duration, years		4.0 ± 2.61	NA	NA
H–Y stages				
H–Y II	7 (29.2%)	3 (25.0%)	4 (33.3%)	0.69
H–Y III	9 (37.5%)	4 (33.3%)	5 (41.7%)	
H–Y IV	8 (33.3%)	5 (41.7%)	3 (25.0%)	
LEDD, mg	598.86 ± 364.99	621.60 ± 368.89	576.13 ± 375.96	0.77
L-dopa	23 (95.8%)	12 (100%)	11 (91.7%)	0.31
D-Agonist	12 (50%)	7 (58.3%)	5 (41.7%)	0.41
MAOBI	3 (16.7%)	1 (8.3%)	2 (16.7%)	0.54
MDS-UPDRS				
Total	61.13 ± 27.92	71.25 ± 34.21	51.0 ± 15.4	0.075
Part I	12.92 ± 6.95	13.92 ± 8.17	11.92 ± 5.66	0.49
Pain and other sensations score				
1 (Slight)		1 (8.3%)	NA	NA
2 (Mild)		8 (55.7%)	NA	NA
3 (Moderate)		2 (16.7%)	NA	NA
4 (Severe)		1 (8.3%)	NA	NA
Part II	16.13 ± 9.76	19.58 ± 11.57	12.67 ± 6.27	0.082
Part III	26.83 ± 13.00	31.58 ± 15.22	22.08 ± 8.5	0.072
Part IV	5.25 ± 3.71	6.17 ± 4.09	4.33 ± 3.20	0.23
MMSE score	24.58 ± 4.14	25.0 ± 3.95	24.17 ± 4.45	0.63
FAB score	12.96 ± 2.56	13.08 ± 3.06	12.83 ± 2.08	0.82

Note: Data are presented as mean ± SD. Disease duration was calculated as the number of years since PD diagnosis. Hoehn and Yahr stages were used to assess the severity of motor symptoms. The Movement Disorder Society Unified Parkinson’s Disease Rating Scale was administered to the patients in the “on” phase. The total dose of medication was converted to a levodopa-equivalent daily dose in mg [16]. PD = Parkinson’s disease; H–Y = Hoehn and Yahr; LEDD = levodopa-equivalent daily dose; L-dopa = L-3, 4-Dihydroxyphenylalanine; D-Agonist = dopamine agonist; MAOBI = monoamine oxidase-B inhibitors; MDS-UPDRS = Movement Disorder Society Unified Parkinson’s Disease Rating Scale; MMSE = Mini-Mental State Examination; FAB = frontal assessment battery; SD = standard deviation; NA = not available.

**Table 2 brainsci-11-01224-t002:** Comparison of the whole-brain and regional coherence values in the alpha band between patients with PD, with and without pain.

	PD with Pain	PD without Pain	*p* Value
Whole brain	0.46 ± 0.12	0.42 ± 0.07	0.39
FF	0.63 ± 0.11	0.59 ± 0.08	0.40
FP	0.45 ± 0.15	0.42 ± 0.07	0.49
FT	0.37 ± 0.12	0.32 ± 0.07	0.16
FO	0.31 ± 0.13	0.27 ± 0.12	0.35
PP	0.58 ± 0.13	0.58 ± 0.08	0.96
PT	0.46 ± 0.12	0.45 ± 0.08	0.77
PO	0.52 ± 0.11	0.52 ± 0.11	0.91
TT	0.44 ± 0.10	0.35 ± 0.08	<0.05
TO	0.44 ± 0.11	0.44 ± 0.11	0.92
OO	0.64 ± 0.10	0.60 ± 0.10	0.35

Note: Data are presented as mean ± SD. FF = frontal–frontal; FP = frontal–parietal; FT = frontal–temporal; FO = frontal–occipital; PP = parietal–parietal; PT = parietal–temporal; PO = parietal–occipital; TT = temporal–temporal; TO = temporal–occipital; OO = occipital–occipital; SD = standard deviation.

## Data Availability

The data presented in this study are available on request from the corresponding author.

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
