# Peer review of "Neuronal Dynamics of Pain in Parkinson’s Disease"

_brainsci, 2021, doi:10.3390/brainsci11091224_

Round 1
Reviewer 1 Report
This is an original research looking into Pain in Parkinson disease. The role of changes in inter-regional cortical synchronization in the pathophysiology of Parkinson's disease and pain has not been very well studied. Possible disruptions in the connectivity can be part of the mechanism of production of pain in affected patients. These disruptions have been investigated here using EEG coherence analysis between set of electrodes in the cortex and comparing pain with no pain PD patients.
I have several points to discuss with the authors that can improve the manuscript in order to help the reader in understand and reproduce the experiments in the paper:
In material and methods:
Line 49 « The exclusion criteria were (1) patients receiving medications .. » : …Please include here the pain medication administrered or the procedures used to alleviate the pain.
Also, How severe was the pain and how was alleviated or not with medication? … , what antiparkinsonian medication?... DBS surgery? Please be more specific
Line 51 … (3) patients with other known causes of pain (e.g., orthopedic disease, peripheral neuropathy, spinal cord diseases…
There is no description of the kind of pain in the pain positive group. The types of pain associated with Parkinson’s include:
- Aching or burning pain from muscles or skeleton,
- Sharp pain from a nerve or nerve root, numbness or “pins and needles”
- Pain also radiating from a nerve or nerve root,
- Pulsing or aching pain that results from tightness or ongoing twisting and writhing movements (dyskinesia), restlessness caused from akathisia and sudden, sharp burning pain that occurs for no known reason.
Also, there should be information about the duration (years, month) of the pain and if it is continuous or discontinuous...There is a lack of epidemiological data of the pain
Line 53 « Pain was assessed using the Movement Disorder Society Unified Parkinson’s Disease Rating Scale Part III (MDS-UPDRS-III) …» UPDRS is not designed to evaluate pain. There are pain specific scales in the literature: In general data on pain in PD has been obtained using nondisease‐specific tools to assess pain, such as the McGill pain scale, the 36‐item Short Form Health Survey, and the Brief Pain Inventory. Recently, the King's Parkinson's Disease Pain Scale (KPPS) was published as the first disease‐specific scale to assess pain in PD. You should include a pain score system
Line 58 « EEG examinations and clinical assessment were performed at the same time of day on separate days within one month, with the patients in the “on” phase... »
Please explain more the conditions of acquisition of the data: When the medication was taken? After how many hours the EEG was done ?. What medication was taken: ( L-dopa ± dopamine agonists) ?
In EEG Analysis
Line 69: “Time-frequency analysis was performed across all 16 electrodes using fast Fourier transformation with a wavelet transform and a frequency resolution of 1 Hz”.
There is no data related to time frequency analysis in the results or in the appendix of the paper
Line-74: “Coherence analysis was performed…”
How the coherence was calculated? Please explain or include a reference here
Line 80: “To evaluate inter-and intra-regional connectivity, regional coherence was compared between the pain-positive and pain-negative groups”
How do you investigate this? This area of the paper need more detailed descriptions of the methods used. The authors should define clearly the different elements (coherence, whole brain coherence, intra-regional connectivity etc...). This part will need more detailed explanation in order to be understandable for the reader.
Line 98 Table 1. Demographic and clinical characteristics of the participants:
Please include here the medication taken by the patient.
Line 125: Table 2. Comparison of the whole-brain and regional coherence values between patients with PD with and without pain.
In which frequency band?
Line 139: “Here, EEG coherence in the alpha frequency band, which is used to evaluate neuronal synchronization in brain networks, was associated with pain modulation”
There is no evidence in the results for this statement: this is the first time that alpha band was mentioned in the paper. No mention of alpha band in the results
Finally, authors should include the limitation of the study in order to achieve the conclusions (i.e. lack of sEEG or invasive data to compare subcortical structures…)
In conclusion, this is an innovative approach but the readers need more data and a better explanation of the analysis in order of fully accept the conclusions of the paper.
Reviewer 2 Report
Comments/queries:
The present brief report entitled " Neuronal dynamics of pain in Parkinson’s disease" by Kinugawa et al. demonstrates the clinical evaluation of pain in sporadic PD patients by using EEG and compared the whole brain and regional coherence data as a single paradigm. However, there are some comments/queries listed below:
Query#1: authors recruited only sporadic PD patients in the present study, how about idiopathic PD cases? Is there any specific rationale to not consider patients with genetic risk factors?
Query#2: Although, authors included late onset PD cases in their study, how about young onset PD cases? Pain is also reported in early onset and as disease progresses. Is there any specific criteria for selecting late onset PD?
Query#3: In Table 1, maximum disease duration is reported not more than 10 years, however in Figure 1, Pearson’s correlation coefficient for disease duration mentioned patients with >10 years. With pain and total, atleast one patient each is ~30 years for disease duration. Is this a typographical error? Please explain.
Query#4: There is a considerable difference in the disease duration of PD with/without pain (Table 1). PD with pain is ~9+/-7.86, while in case of PD without pain is ~6.5+/-3.66. This could be a big confounding factor in regard to pain diagnosis. PD without pain patients need to be age matched to PD with pain patients to support the data on EEG coherence.
Query#5: Number of sporadic PD patients recruited for the study is very less which is a major confounding factor to conclude the findings.
Line#95: “statistically significant between-group differences” needs to be re-written.
Round 2
Reviewer 1 Report
Authors have answer all the points in my previous review. The only remaining point, in my view, is that as this was a single variable investigation (EEG connectivity), the authors should be cautious when putting too much emphasis in this as solely mechanism of pain in sporadic PD (in the conclusion "...EEG analysis sheds light into the pathological mechanisms underpinning of pain in PD...").
Reviewer 2 Report
The present manuscript by Kinugawa et al. has been revised point wise with by citing relevant literature. Authors have tried to revise their manuscript by answering some queries that were of concern to support their findings. However, considering the sample size too small in their study and greater degree of variability in disease duration and age, I recommend authors must increase the sample size and age matched to make this study deliverable, otherwise the study is inconclusive in its current stage.
